# The Use and Performance of Artificial Intelligence in Prosthodontics: A Systematic Review

**DOI:** 10.3390/s21196628

**Published:** 2021-10-05

**Authors:** Selina A. Bernauer, Nicola U. Zitzmann, Tim Joda

**Affiliations:** UZB University Center for Dental Medicine Basel, Department of Reconstructive Dentistry, University of Basel, 4058 Basel, Switzerland; selina.bernauer@unibas.ch (S.A.B.); n.zitzmann@unibas.ch (N.U.Z.)

**Keywords:** artificial intelligence, machine learning, deep learning, neural networks, reconstructive dentistry, prosthetic treatment

## Abstract

(1) Background: The rapid pace of digital development in everyday life is also reflected in dentistry, including the emergence of the first systems based on artificial intelligence (AI). This systematic review focused on the recent scientific literature and provides an overview of the application of AI in the dental discipline of prosthodontics. (2) Method: According to a modified PICO-strategy, an electronic (MEDLINE, EMBASE, CENTRAL) and manual search up to 30 June 2021 was carried out for the literature published in the last five years reporting the use of AI in the field of prosthodontics. (3) Results: 560 titles were screened, of which 30 abstracts and 16 full texts were selected for further review. Seven studies met the inclusion criteria and were analyzed. Most of the identified studies reported the training and application of an AI system (*n* = 6) or explored the function of an intrinsic AI system in a CAD software (*n* = 1). (4) Conclusions: While the number of included studies reporting the use of AI was relatively low, the summary of the obtained findings by the included studies represents the latest AI developments in prosthodontics demonstrating its application for automated diagnostics, as a predictive measure, and as a classification or identification tool. In the future, AI technologies will likely be used for collecting, processing, and organizing patient-related datasets to provide patient-centered, individualized dental treatment.

## 1. Introduction

“Digital transformations”, “digitized workflows”, “technical developments”: these terms describe some of the game changers of the 21st century, both in social life as well as dental medicine [1]. The use of mobile devices, tablets, and smartphones, and the easy access to technology and the World Wide Web, have changed the cultural habits of our society in general [1]. It is not surprising that more advanced technologies, such as artificial intelligence (AI), are also finding increasing application in daily life. Generally speaking, AI imitates the cognitive processes of human intelligence by using machines and software-type algorithms to manage complex tasks [2,3].

AI applications are commonplace in digital everyday life, for example in the form of virtual assistants such as “Siri” or “Alexa” [1,4], and AI applications are implemented in various engineering fields [5,6,7]. In medicine, AI algorithms are ubiquitously used for image processing via feature extraction of specific images and target conducting [8]. For example, AI algorithms can be used for analyzation of chest X-ray and lung CT image samples images of COVID-19 patients and thus decrease the picture diagnostic time of the radiologist and accelerate clinical decisions [9]. Furthermore, technological advances are increasingly noticeable in dentistry, but are still restrained compared to medicine. Digital dental processes are continuously being standardized and are becoming part of routine treatment protocols [10,11,12]. In particular, computer-aided design/computer-aided manufacturing (CAD/CAM) procedures have found their way into everyday clinical and laboratory practice [10,13]. Digitization in dentistry continues to develop and new trends are emerging, including the application of AI.

AI is a widely used term that warrants a precise definition (Figure 1); in general, algorithms that simulate the processes of human intelligence to resolve problems are summarized under the term AI [4,14]. Intelligence is defined as: “the ability of a system to act appropriately in an uncertain environment, where appropriate action is that which increases the probability of success, and success is the achievement of behavioral subgoals that support the system’s ultimate goal” [15]. Strong and weak AI can be differentiated. Strong AI describes a system “that was operated in the same way as human intelligence through non-natural, artificial hardware, and software construction” [4]. Weak AI, on the other hand, does not aim to mimic human intelligence in its entirety, but rather is a system, in which the advantages of medical and logical algorithms can be used by humans. Weak AI recognizes that computer-integrated technology is different from human intellectual performances [4].

Inspiration for the formation of artificial networks can be found in the biology of the human brain. “An artificial neuron is a mathematical function conceived as a coarse model of a biological neuron. The principle is to simulate the transfer of information through a neuron: weighted nodes receive the inputs (representing the synapses), sum them to produce an activation (representing the axon), and pass this activation to a nonlinear function called activation or transfer function, in order to generate the output signal. Each neuron acts as an elementary processing unit. The output signal of one unit will feed the other units, organized in layers, and so on, forming an artificial neural network (ANN)” [16]. There are typically three phases in the development and application of an ANN: (a) the training phase, where the ANN learns; (b) the validation phase, where the ANN’s results and reference values are reconciled; and (c) the inference or application phase, where the ANN is used in real cases [14].

Machine learning (ML) is part of weak AI; by recognizing patterns, the computer system is able to learn and make predictions [4,17]. After being trained on existing datasets, the ML system is able to provide accurate predictions on new data. Often the training datasets need to be simplified, so that the algorithm can recognize the desired pattern. Depending on how carefully these interventions are made, the performance of ML systems varies [16]. Current examples of ML systems include translators such as DeepL, diagnosis tools, and speech recognition frameworks [18]. Deep learning (DL) is part of ML that describes “a set of computational models composed of multiple layers of data processing, which make it possible to learn by representing these data through several levels of abstraction” [16]. Compared to ML, deep neural networks (DNNs) are able to independently learn and hierarchize the training dataset but require, therefore, larger amounts of data. It comes with the advantage that new feature extractors are not needed for every problem [16]. DL works via a backpropagation algorithm to train neural networks [16]. Convolutional neural networks (CNN) are a special part of DL, used especially for image processing and analyses of radiological datasets [16].

AI technologies are well suited for repetitive work using large datasets. The greater the workload, the more precise the performance of AI becomes [1]. Since dentistry provides many areas for assisting work and automation of simple routine tasks, AI could find applications here and support dentists to improve quality of work and accuracy [18].

The domain of AI in (dental) medicine is image data processing in radiological workflows, especially for detection of caries and periapical endodontic lesions [16], as well as automatic identification and classification of oral implant systems [19]. In prosthodontics, however, AI technology is still scarce due to complex diagnostics and treatment requiring individual protocols.

Therefore, the aim of this systematic review was to analyze the recent scientific literature on the diagnostic performance and the clinical applications of AI in applied prosthodontics. The remainder of this systematic review was structured as follows: (a) introduction of the search strategy and related inclusion criteria for data extraction; (b) summary of the obtained findings of the identified and included studies; (c) critical discussion of the results; (d) conclusions.

## 2. Materials and Methods

This systematic review was conducted in accordance with the guidelines of PRISMA (Preferred Reporting Items of Systematic Reviews and Meta-Analyses) [20].

### 2.1. Search Strategy

Based on the PICO criteria, a search strategy was developed and an electronic search was conducted. The PICO question was formulated as follows: What are the current clinical applications and diagnostic performance of AI in the field of prosthodontics?

A systematic electronic search of PubMed MEDLINE, EMBASE, and CENTRAL was performed for English-language publications; in addition, Google Scholar was reviewed. Search syntax was categorized into population, intervention, comparison, and outcome (PICO). Each category was assembled from a combination of Medical Subject Headings (MeSH Terms) as well as free-text words in simple or multiple conjunctions: ((Prosthodontics [Mesh]) OR (prosthetic treatment) OR (reconstructive therapy)) AND ((Artificial Intelligence [Mesh]) OR (Machine Learning OR Deep Learning OR Neural Networks [Mesh])) (Table 1).

Additional manual searches of the bibliographies of all full-text articles and related reviews selected from the electronic search were also performed. Furthermore, manual searching was conducted in the following journals: Journal of Prosthodontic Research, Journal of Prosthetic Dentistry, Clinical Oral Implants Research, International Journal of Oral Maxillofacial Implants, Clinical Implant Dentistry and Related Research, Implant Dentistry, and Journal of Implantology.

### 2.2. Inclusion and Exclusion Criteria

Inclusion criteria for the studies were defined as follows:Studies at all levels of evidence, except expert opinion;Articles published in English;Articles published in the last 5 years (up to 30 June 2021).

Exclusion criteria for the studies were defined as follows:Review articles, letter to editors and case reports involving less than 5 cases;Animal studies;Full-text not available/accessible.

### 2.3. Data Extraction

Two reviewers (S.B. and T.J.) independently screened the titles and abstracts that were identified in the searches according to the defined inclusion and exclusion criteria. If sufficient information could not be extracted from titles and abstracts, the full text was consulted. The full texts of potentially relevant articles were obtained and reviewed in detail by both reviewers, from which the final list of articles was selected for further analysis. Disagreements were resolved by discussion.

The following information was collected from selected articles:Author(s), year of publication, country, study design;Total number of patients/datasets;Training/validation datasets;Test datasets;Aim of the study;AI application; andOutcome.

The information extracted from the articles was tabulated. Assessment of risk of bias was carried out for each included study, using the Newcastle–Ottawa Assessment Scale (http://www.ohri.ca/programs/clinical_epidemiology/oxford.asp (Accessed on 7 September 2021)) to assess the quality of the studies (Table 2).

## 3. Results

### 3.1. Included Studies

The systematic search was finished on 30 June 2021. A total of 560 article titles were screened and 30 abstracts were considered for further analysis. Subsequently, the full texts of 16 articles were analyzed to determine if they met the inclusion criteria. After further examination, 11 articles were excluded due to the following reasons:Not a study in the field of AI application in prosthodontics (*n* = 6);Not a clinical study (*n* = 3);Full text not available (*n* = 1);Missing information on AI technology (*n* = 1).

Moreover, two other articles meeting the inclusion criteria were added from manual searching; subsequently, a total of seven full-text articles were included for data extraction (Figure 1).

### 3.2. Descriptive Analysis

The seven included full-text articles came from diverse areas of the field of prosthodontics and demonstrated a wide range of the use of AI. Six studies investigated the training and application of different AI systems [19,22,23,24,25,26] and one study explored the function of an intrinsic AI system of a specific CAD software for designing prosthetic reconstructions [21]. Direct comparisons of the identified and included studies were not feasible because of the heterogeneity of the specific aims, defined outcomes, and topics within the field of reconstructive dentistry. Therefore, the analysis of the information reported by the included studies follows a descriptive approach. The detailed data extraction is summarized in Table 3.

All included studies were designed as non-randomized, retrospective cohort studies. Three of the publications were from one research group with a special focus on radiographic image analysis [19,23,24]. The number of datasets used for AI training in the studies varied considerably (from 43 to 10770). Five of the studies included investigated CNN models, one study applied an ANN model, and another study used an intrinsic AI and algorithms of a commercially available laboratory CAD software.

## 4. Discussion

This systematic review focused on applied AI technologies in prosthodontics, and it was demonstrated that AI was used for automated diagnostics, as a predictive measure, and as a classification or identification tool. The findings indicated that, in the wider field of prosthodontics, AI has been applied to CAD/CAM systems, implant prosthetics, tooth preservation, and orofacial anatomy. Since digital technologies are developing rapidly and the observed turnover rate of obsolete software is about 1.5–2 years, the current search was limited to the last 5 years. The aim of this systematic review was not to provide a historical overview of AI technology in dentistry in general. Rather, the manuscript focused on prosthetic AI applications. Furthermore, the systematic search focused on clinical trials and case series with at least 5 patients to increase the scientific level (and to avoid including technical reports).

The overall number of eligible studies investigating AI applications in prosthodontics was relatively low (*n* = 7). Although automated detection of caries has been investigated in AI dental imaging diagnostics a few years ago, the use of AI technology in prosthodontics is (still) rare. Prosthodontics itself is a diverse and complex area of dental medicine and one that may benefit from the routine application of AI technologies. Long-term success relies on good prognosis of abutment teeth (and/or implants) in all facets of periodontal, endodontic, operative and reconstructive principles, and include patient-specific factors, such as load situation, personal and medical issues, and supportive care [27]. Successful prosthetic reconstructions require the use of a synoptic treatment concept with sufficient backward planning and a clean practical implementation, including dental laboratory workflows. Furthermore, the various different patient-related factors must be taken into account; many intraoral situations are not directly comparable, with the exception of edentulous patients. One factor is that the number of teeth to be replaced varies extremely from 1 to 28. Mathematically, the number of possible situations, starting from 28 teeth, would be the factorial of 27, i.e., 27×, 26×, 25×, 24×, etc. The number of options, combined with the factors mentioned above, makes prosthetic therapy a demanding and complex specialty, which requires a certain knowledge in adjacent disciplines. AI technologies are particularly suited to dealing with complex situations with multiple possible factors; therefore, the application of AI to prosthetic workflows is of high interest.

This systematic review revealed that AI systems are currently mainly limited to test versions of automated diagnostics, especially in dental imaging and radiology [3], and classification tools such as for periodontally compromised teeth, dental cusps, or caries. Encouragingly, the identified studies all reported high performance of the different AI systems investigated (Table 3), i.e., including high diagnostic accuracy for dental caries [23], and very good prediction accuracy for debonding of CAD/CAM crowns [22].

AI is increasingly being applied to dentistry including in diverse areas of prosthetic research for efficient data processing. The first application of AI within dentistry was in the classification of diagnostic images and processing of data from surface scanning techniques, because the digitally coded images could easily be transferred into AI systems [3,17]. The application of AI in diagnostics continues to be developed. AI technologies in dentistry have the power to become central in the triad of patient data management, health care application, and services, and can facilitate future developments in patient-centered individualized treatment [3].

Beyond prosthodontics, AI was previously linked to other dental disciplines [2]. In tooth preservation, radiologically driven AI analyses can help detect root fractures and identify periapical pathologies [28] or classify root morphologies [29]. In periodontology, disease progression can be evaluated while clinical and radiological periodontal parameters are automatically determined following AI technology. In oral surgery, AI can be used to screen radiological images for pathological changes, such as cysts and bony tumors [2]. Furthermore, there are possible applications in implantology. AI-based treatment planning in CAD/CAM implant dentistry could be of great interest in order to simplify virtual 3D treatment planning, and, consecutively, robotic insertion of dental implants using AI applications [30].

AI will certainly play a significant role in dentistry in the future and the development of these technologies is awaited with excitement. AI has a disruptive potential to renew processes in all fields of dentistry; but, due to the complexity of prosthetic treatment concepts, the adoption of AI technology in prosthodontics is still rather hesitant. AI systems are particularly beneficial for processing and analyzing large amounts of data to classify outcomes, and for processing repetitive workflows. AI algorithms will likely provide support in evidence-based dental decision making, particularly for less experienced practitioners, and facilitate the analysis of individual patient cases. More homogeneous treatment protocols could be ensured that still allow for individualized and personalized treatment. Chen et al. developed a prototype decision model to assist (unexperienced) dentists in choosing appropriate removable prosthetic options [31]. Such supportive tools in therapy planning of (complex) patient cases in prosthodontics enable the further development of tele-dentistry.

The combination of AI technologies in the field of prosthodontics could lead to a wide variety of novel options, such as AI systems for generating occlusal surface design for crowns accounting for existing intraoral wear facets, as automatic set-up designers for complete dentures, for determining the emergence profile in implantology, or in automatic framework designs for removable partial dentures [32]. Finally, as an educational tool, AI already provides the opportunity to support less experienced undergraduate students in their professional development [33].

In principle, there are no limitations to using the power of AI in prosthodontics. In many cases, however, it is (unfortunately) financial factors that trigger the development and standard implementation of new AI technologies in the most economically profitable areas—and tend to prevent interesting dental applications because a market appears too small for the dental industry. Nevertheless, future developments and research on AI can be eagerly awaited.

Although this seems to be very promising, the obstacles of AI systems should not be ignored. Based on this systematic literature search, the identified trials must be seen as pilot studies and experimental, and the technologies used in these studies are not yet suitable for everyday clinical routine in the focus of prosthodontics. At the moment, AI is mainly used in undergraduate dental education and for academic research. Improvements in the technology and user interfaces is needed before the techniques can be implemented into routine dental practice. Some intermediate steps still need to be developed to make AI attractive and helpful for its routine implementation. Beyond the technical possibilities, the cost–benefit ratio and ethical aspects of the use of patient-specific data also need to be critically examined in future research on AI technology in (reconstructive) dentistry [3].

## 5. Conclusions

While the number of studies reporting the use of AI in prosthodontics identified in this systematic review was relatively low (inclusion of seven out of sixteen full-texts), these studies reflect an honest overview of the latest developments in AI focusing on diagnostics, predictive measures, classification, and identification tools. Prosthodontics is the dental discipline with the greatest variance in terms of diagnostics and, in particular, treatment options. Therefore, it does not seem surprising that the results of this systematic review are comparably diverse and heterogeneous.

In addition, dentistry is (still) lagging behind medicine in terms of routine use of AI technology. Nevertheless, AI applications in prosthodontics have the potential to open up a wide range of opportunities for clinicians and patients, and could be used as a supplementary future basic tool for collecting, processing, and organizing patient-related datasets to provide patient-centered, individualized, and personalized treatment. Possible applications of AI algorithms are very diverse in prosthodontics and one can eagerly await further research foci and developments.

The step-by-step implementation of digital applications (in dentistry) is linked to the necessary technical development and is dependent on the flexibility of the users in everyday life. The willingness to learn new treatment protocols and to trust computerized applications has been proven as a strong negative driver for dentists [34]. In addition, new technologies require continuous investment of the dental community. As long as the research results are not demonstrating superiority, the routine implementation of AI applications in prosthodontics will be delayed. Here, the university and dental schools need to foster AI technology in research and education.

A critical topic and crucial factor for the successful rehabilitation of complex prosthetic cases is the correct clinical definition of vertical and horizontal maxillo-mandibular relationships. AI technology could be used for automatic registering of jaw relationships based on radiological landmarks in cone-beam computed tomography. In this context, the configuration of virtual dental articulators could be synchronized with the radiological situation to simulate individual patient movements for treatment simulation and final fabrication of prosthetic reconstructions.

The use of AI technologies in prosthodontics is conceivable in many ways (and absolutely desirable) from the clinicians’ point of view, but not least the demand and the economic efficiency will decide whether the MedTech industry will push AI in prosthodontics as fast and to what extent.

## Figures and Tables

**Figure 1 sensors-21-06628-f001:**
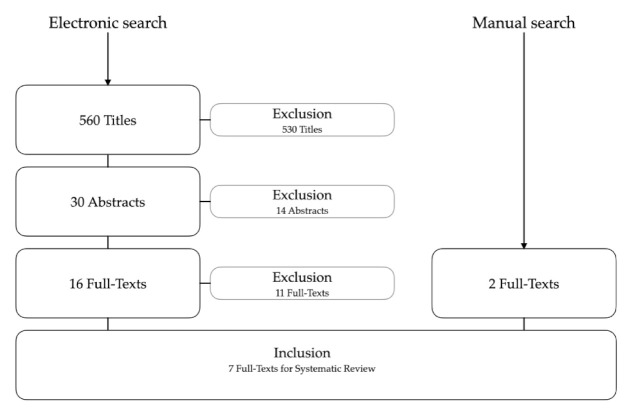
Flowchart of the electronic and manual search strategy.

**Table 1 sensors-21-06628-t001:** Search strategy according to the PICO criteria.

Focused Question (PICO)	What Are the Current Clinical Applications and Diagnostic Performance of Artificial Intelligence (AI) in Prosthodontics?	
Search Strategy	Population	Patients with indication for prosthetic reconstructions #1 ((Prosthodontics [Mesh]) OR (prosthetic treatment) OR (reconstructive therapy))
	Intervention or exposure	Diagnostic model based on applied AI algorithms #2 ((Artificial Intelligence [Mesh]) OR (Machine Learning OR Deep Learning OR Neural Networks [Mesh]))
	Comparison	N.A.
	Outcome	Clinical applications or diagnostic performance of the proposed AI model
	Search combination	#1 AND #2 Limitations: Articles published in the last 5 years (up to 30 June 2021); English
Database search	Electronic	PubMed Medline, Embase, Central, manual search
	Journals	Journal of Prosthodontic Research, Journal of Prosthetic Dentistry, Clinical Oral Implants Research, International Journal of Oral Maxillofacial Implants, Clinical Implant Dentistry and Related Research, Implant Dentistry, Journal of Implantology
Selection criteria	Inclusion criteria	Studies at all levels of evidence, except expert opinion; Articles published in English;Articles published in the last 5 years.
	Exclusion criteria	Review articles, letter to editors and case reports/case series involving less than 5 cases; Animal studies; Multiple publications on the same patient population; Full text not available/accessible.

**Table 2 sensors-21-06628-t002:** Presentation of risk of bias evaluation for included studies.

	Selection (Max. 4 Stars)	Comparability (Max. 2 Stars)	Outcome (Max. 4 Stars)
Lee, J.H. et al. (2020)	**	−	*
Lerner, H. et al. (2020)	***	−	*
Yamaguchi, S. et al. (2019)	***	*	*
Lee, J.H. et al. (2018a)	**	−	*
Lee, J.H. et al. (2018b)	**	−	*
Raith, S. et al. (2017)	**	−	*
Wei, J. et al. (2016)	*	−	*

**Table 3 sensors-21-06628-t003:** Characteristics and outcomes of the studies included.

First Author (Year) Country	Study Design	n Datasets	Training/ Validation Datasets	Test Datasets	Aim of the Study	AI Application	Outcome
Lee (2020) [19] Korea	Retrospective cohort study	10,770 radiographic images	6462 (60%) 2154 (20%)	2154 (20%)	“The aim of the current study was to evaluate the efficacy of deep CNN algorithm for the identification and classification of dental implant systems”.	CNN (GoogLeNetInception v3)	“Deep CNN architecture is useful for the identification and classification of dental implant systems using panoramic and periapical radiographic images”.
Lerner (2020) [21] Germany	Retrospective cohort study	106 restorations	n.r.	n.r.	“Purpose of this retrospective clinical study is to present a protocol for the use of AI to fabricate implant-supported monolithic zirconia crowns cemented on customized hybrid abutments, via a full digital workflow”.	Intrinsic AI and algorithms of the CAD software (Valletta^®^, Exocad, Darmstadt, Germany)	“Using intrinsic AI, the software was able to automatically trace the margin line of the implant abutment, though subgingival”. In 96.2% of the restorations, the marginal adaption was very accurate.
Yamaguchi (2019) [22] Japan	Retrospective cohort study	8640	6480	2160	“The aim of this study was to assess the validity of deep learning with a CNN method to predict the debonding probability of CAD/CAM composite resins restorations from 2D images captured from 3D STL models of a die scanned by a 3D oral scanner”.	CNN; implemented with the Keras library (version 2.2.4) on top of TensorFlow (GPU version 1.12.2) in Python (version 3.7.2)	High performance of AI in predicting the debonding probability of 2160 test 2D-images of CAD/CAM crowns with a current prediction accuracy of 98.5%”.
Lee (2018a) [23] Korea	Retrospective cohort study	3000 periapical radiographic images	2400 (80%)	600 (20%)	“The aim of the current study was to evaluate the efficacy of deep CNN algorithms for detection and diagnosis of dental caries on periapical radiographs”.	CNN (GoogLeNetInception v3)	High diagnostic accuracies of premolar (89%), molar (88%) and both premolar and molar (82%) models were achieved. “Deep CNN algorithms are expected to be among the most effective and efficient methods for diagnosing dental caries”.
Lee (2018b) [24] Korea	Retrospective cohort study	1740 periapical radiographic images	1044	348	“The aim of the current study was to develop a computer-assisted detection system based on a deep CNN algorithm and to evaluate the potential usefulness and accuracy of this system for the diagnosis and prediction of periodontally compromised teeth”.	CNN; based on a Keras framework in Python	“With the deep learning algorithm, the diagnostic accuracy for periodontally compromised teeth was 81.0% for premolars and 76.7% for molars. […] The deep CNN algorithm was useful for assessing the diagnosis and predictability of PCT”.
Raith (2016) [25] Germany	Retrospective cohort study	129 datasets	n.r.	n.r.	“[The] hypothesis is that tooth classification algorithms based on ANNs are capable of classifying teeth with sufficient accuracy for potential use in clinical practice in order to improve digital workflow in dental prosthetics”.	ANN; principal algorithm based on blob detection with a Difference of Gaussians (DoG) approach, implemented in Python programming language	High performance with correct classifications were shown; “cusps are detected automatically and thus completely reproducible, which is advantageous when standardized treatment concepts need to be established, paving the way for evidence-based dentistry”.
Wei (2016) [26] China	Retrospective cohort study	43 datasets	39	4	“[The aim of this study was] to explore the feasibility of a novel computer color-matching system based on the improved back-propagation neural network model by comparing it with the traditional visual method”.	CNN; back-propagation neural network (BPNN) is a multilayer feed-forward neural network trained by the error back-propagation algorithms	“The novel computer color matching system produced greater accuracy in color reproduction within the given color space than the traditional visual approach”.

AI: artificial intelligence; ANN: artificial neural network; CNN: convolutional neural networks; n: number; n.r.: not reported.

## Data Availability

The data are not publicly available due to privacy restrictions.

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
