# Peer review of "The Use and Performance of Artificial Intelligence in Prosthodontics: A Systematic Review"

_sensors, 2021, doi:10.3390/s21196628_

Round 1
Reviewer 1 Report
The review is devoted to the application of artificial intelligence in prosthodontics. The topic is interesting and actual. Undoubtedly, the use and development of AI methods in medicine, clinical research and, of course, prosthodontics can allow us to increase the effectiveness of both the diagnostic and treatment of appropriate disease. However, to my mind, the presented paper has some significant shortcomings which should be corrected. Below, I present my remarks.
- In my view, the structure of the review is distinguished from the research article. You do not obtain the results, you analyse the current state of research in this subject area with allocation their advantages and shortcomings. So, the sections Results and Discussion do not correspond to this type of publication. To my mind, the manuscript should be reformatted.
- Moreover, the review is very short. It has not covered such areas of AI application as Image recognizing with diagnoses classification using deep learning techniques.
I think, that in this form the manuscript cannot be accepted.
Author Response
Reviewer: 1
Comments to the Author
The review is devoted to the application of artificial intelligence in prosthodontics. The topic is interesting and actual. Undoubtedly, the use and development of AI methods in medicine, clinical research and, of course, prosthodontics can allow us to increase the effectiveness of both the diagnostic and treatment of appropriate disease. However, to my mind, the presented paper has some significant shortcomings which should be corrected. Below, I present my remarks.
RESPONSE
Thank you very much for your time to review the manuscript and for your help in improving the quality of the manuscript.
In my view, the structure of the review is distinguished from the research article. You do not obtain the results, you analyse the current state of research in this subject area with allocation their advantages and shortcomings. So, the sections Results and Discussion do not correspond to this type of publication. To my mind, the manuscript should be reformatted.
RESPONSE
Thank you very much. You are absolutely right. We clarified the language of the section in the results section: “A direct comparison of the identified and included studies was not feasible because of the heterogeneity of the specific aims, defined outcomes, and topics within the field of reconstructive dentistry. Therefore, the analysis of the information reported by the included studies follows a descriptive approach. The detailed data extraction of those studies is summarized in Table 2.”
Moreover, the review is very short. It has not covered such areas of AI application as Image recognizing with diagnoses classification using deep learning techniques.
RESPONSE
The focus of this Systematic Review was on AI technology and applications in prosthodontics. In contrast to medicine, AI technology is generally still used rather rarely in dentistry. Radiological imaging is an important field of interest for AI-based diagnostics. In general dentistry, the detection of caries and endodontic lesions are of increasing interest. However, in prosthodontics, diagnostics and treatment (planning) are very complex and require an individual approach. Therefore, the number of identified studies was low(er) at this time. We wanted to analyze the current status of AI applications in prosthodontics and to stimulate the discussion of this technology in this special dental discipline.
Reviewer 2 Report
Authors conducted this research in the title of "The use and performance of artificial intelligence in prosthodontics: A systematic review".
The paper’s subject could be interesting for readers of journal. Therefore, I recommend this paper for publication in this journal but before that, I have a few comments on the text that should be addressed before publication:
Comments:
1) Introduction: Most of the introduction section is allocated to history and general terms of Artificial neural networks. The connection between prosthodontics and artificial neural networks should be emphasised in this section. In other words, the applications of artificial neural networks in prosthodontics should be more bold.
2) About Figure 2: The figure 2 is a table, but authors considered it as figure! It should be corrected by authors.
3) Figure 2: The title of figure 2 is too long and unclear.
4) The mentioned studies in figure 2 have not been ordered by year correctly. It should be from older to newer or on the contrary.
5) Some used fonts in table 1 are too small and it is hard to read them easily.
6) Authors did not mention much things about used softwares in related and similar articles and works in this research.
7) The conclusion section is too short and unclear. Authors should explain more about their suggestions for future works.
8) In figure 3: The color of used fonts at boxes in the middle of figure 3 (like Exclusion 530 titles) is not appropriate. It makes it hard to read it easily.
9) Since recently it has been proved that artificial neural network (ANN) would have a numerous applications in all of engineering fields, I highly recommend the authors to add some references in this manuscript in this regard. It would be useful for the readers of journal to get familiar with the application of ANN in other engineering fields. I recommend the others to add all the following references, which are the newest references in this field.
[1] Moradi, M. J., & Hariri-Ardebili, M. A. (2019). Developing a library of shear walls database and the neural network based predictive meta-model. Applied Sciences, 9(12), 2562.
[2] Ganguly, B., Chaudhuri, S., Biswas, S., Dey, D., Munshi, S., Chatterjee, B., Dalai, S. and Chakravorti, S., 2020. Wavelet Kernel-Based Convolutional Neural Network for Localization of Partial Discharge Sources Within a Power Apparatus. IEEE Transactions on Industrial Informatics, 17(3), pp.1831-1841.
[3] Roshani, M.,, et al., 2021. Evaluation of flow pattern recognition and void fraction measurement in two phase flow independent of oil pipeline’s scale layer thickness. Alexandria Engineering Journal.
Author Response
Reviewer: 2
Comments to the Author
Authors conducted this research in the title of "The use and performance of artificial intelligence in prosthodontics: A systematic review".
The paper’s subject could be interesting for readers of journal. Therefore, I recommend this paper for publication in this journal but before that, I have a few comments on the text that should be addressed before publication:
RESPONSE
Thank you very much for your time to review the revised manuscript and for your valuable advices.
1) Introduction: Most of the introduction section is allocated to history and general terms of Artificial neural networks. The connection between prosthodontics and artificial neural networks should be emphasised in this section. In other words, the applications of artificial neural networks in prosthodontics should be more bold.
RESPONSE
Thank you for this important input, we emphasized this connection more clearly; even though the use of AI applications in prosthodontics is still reserved. Beyond that, we also included this aspect in the Discussion as a suggestion for the future.
2) About Figure 2: The figure 2 is a table, but authors considered it as figure! It should be corrected by authors.
RESPONSE
Thank you very much. We changed the term accordingly.
3) Figure 2: The title of figure 2 is too long and unclear.
RESPONSE
The title has been shortened and adapted accordingly.
4) The mentioned studies in figure 2 have not been ordered by year correctly. It should be from older to newer or on the contrary.
RESPONSE
Many thanks for the comment, the publications have been ordered by date of publication (Tables 2 and 3). Now, the layout and format are more reader friendly.
5) Some used fonts in table 1 are too small and it is hard to read them easily.
RESPONSE
The layout has been adjusted as you suggested (Table 1).
6) Authors did not mention much things about used softwares in related and similar articles and works in this research.
RESPONSE
Thank you very much for his valuable comment. Table 3 has been updated and the used software was listed.
7) The conclusion section is too short and unclear. Authors should explain more about their suggestions for future works.
RESPONSE
Thank you very much for this input. We have added more details about future approaches and expanded the Conclusion (Text change: Section “Conclusion”).
8) In figure 3: The color of used fonts at boxes in the middle of figure 3 (like Exclusion 530 titles) is not appropriate. It makes it hard to read it easily.
RESPONSE
The layout has been adjusted accordingly (now Figure 1 – formerly Figure 3).
9) Since recently it has been proved that artificial neural network (ANN) would have a numerous applications in all of engineering fields, I highly recommend the authors to add some references in this manuscript in this regard. It would be useful for the readers of journal to get familiar with the application of ANN in other engineering fields. I recommend the others to add all the following references, which are the newest references in this field.
[1] Moradi, M. J., & Hariri-Ardebili, M. A. (2019). Developing a library of shear walls database and the neural network based predictive meta-model. Applied Sciences, 9(12), 2562.
[2] Ganguly, B., Chaudhuri, S., Biswas, S., Dey, D., Munshi, S., Chatterjee, B., Dalai, S. and Chakravorti, S., 2020. Wavelet Kernel-Based Convolutional Neural Network for Localization of Partial Discharge Sources Within a Power Apparatus. IEEE Transactions on Industrial Informatics, 17(3), pp.1831-1841.
[3] Roshani, M.,, et al., 2021. Evaluation of flow pattern recognition and void fraction measurement in two phase flow independent of oil pipeline’s scale layer thickness. Alexandria Engineering Journal.
RESPONSE
Thank you very much for listing these publications, we are happy to point our readers to further fields of application of AI (Text change: Section “Discussion”).
Reviewer 3 Report
Please see the file attached.

Author Response
Reviewer: 3
Comments to the Author
The study classified the clinical applications and diagnostic performance of AI and machine learning based techniques to Prosthodontics, the updates included and filtered recent study in the last 5 years (till June 30, 2021). It is a concise survey paper oriented on the recent advances in AI applications to Prosthodontics.
The topic is relevant because the research topic is both special and technical orientations are refresh. The figures and tables display the proposed workflow, related keynote work and the specific pointers in a narrow branch of AI applications in medical domain.
Subject area compared with other published material:
It is an integrated systematic review paper, combining the highly condensed results from other published research article especially in the latest 3-4 years (2018 ~ Jun. 2021), the Tables presents the characteristics and outcomes of included study in classified categories, and the Appendix shows their focused questions on search strategy, selection criteria, database, etc.
RESPONSE
Thank you very much for your time to review the manuscript and to help improve the quality.
Specific improvements regarding the methodology
There are a few aspects need to be addressed in the revised version. Regarding the major parts, the only kernel component of the methodlogy is located in Table 1 (Characteristics and Outcomes of the included study), while this table also has quite a few problems, i.e., missing any research outputs from Year 2017, all the technical approaches (Applied AI) are CNN / ANN, and more than half of the results are "not reported"; besides, several set of evaluation parameters (study design, patients, training and test datasets) actually makes limited sense to the research topic itself. As a result, the specific improvements must be addressed to expanding the methodology on (AI-based techniques, case studies, recent progress and applications) to a much large set of evaluation rubrics, which can be convincible and reliable to reviewers and editors.
On the other hand, I think Section 2 and Section 3 both looks generic, which lacks expected professional case study or cited quantitative results from keynote research publications, which undermines the overall impression from reviewers and hinders the current shape to qualify a good systematic review paper.
RESPONSE
Thank you very much for the important inputs to the methodology. Formerly Table 1 (now it is Table 3) has been modified and as much details as given in the full-texts regarding AI have been added, and the mentioned evaluation parameters have been removed as suggested.
The systematic search strategy followed the PICO approach; and consecutively, the manuscript followed the PRISMA guideline. The search yielded 7 hits including extraction of data from these included studies. Consequently, in 2017, we could not find any study that met the inclusion criteria. As only publications from the field of reconstructive dentistry were included in this Systematic Review, it is quite possible that there is no paper from 2017. The aim of this Review was not to write an overview on AI in general dentistry, but very specifically in the field of prosthodontics. However, we considered further publications in dentistry from this period 2016-2021 and mentioned them in the Discussion.
Minor issues to be fixed
Minor issues to be fixed: a) replace "30.06.2021" to "June 30, 2021"; b) make the required notations of "n" italic: i.e., "n = 6", "n = 3", "n = 1", etc; c) stop using hyphens "-" to connect broken words between two adjacent lines. Formatting issues can be adjusted if you using a Word template.
RESPONSE
Thanks again for your effort to improve the quality and style of the manuscript. We addressed all minor issues as suggested.
Conclusion
Conclusions are consistent with the major content, while only two sentences contribute to the conclusion section.
I don't think the address the posed main question. The first sentence only provided a self-evaluation on their study (and claimed to be "good overview"), which is too generic to be accepted by scientific researchers, the second sentence claimed AI technologies to be future basic tool, which is superficial (since AI already had a wide range of applications in medical domain and even the field of Prosthodontics already appeared with more than 500 research articles). In general, current conclusion section is also generic, One successful sample conclusion should include main summary of work, limitations on study, the opening questions, and proposed future study or suggested research orientations, which is basically 2-3 paragraphs, including a lot more crucial statements, highly condensed while without loss of specific points.
RESPONSE
Thank you very much to help improve the Conclusions. We expanded the Conclusions and included aspects as limitations, remaining questions, and future trials.
References
The authors cited 24 references. I think the current status of citations are still weak. The problematic issues include the following parts: a) Missing any relevant work in Years 2011-2015, while only selected keynote research study in the latest 5 years are included; I think both parts should be strengthened. b) Missing milestone work and study in AI and deep learning based models proposed for Prosthodontics (in the most recent few years, 2018-2021). c) Ref. [8] is too old, what are the progressive studys posterior to this paper?
- d) Again, the authors may consider citing more MDPI affiliated journal publicationss in Years 2016- 2021 in medical domain with a focus on AI techniques for dental care and Prosthodontics.
RESPONSE
Thank you very much for your valuable advices and comments. The aim of this Systematic Review was to highlight and focus on the current literature in the last 5 years in prosthodontics and AI applications – as stated in the section M&M: “Since digital technologies are developing rapidly and the observed turnover rate of obsolete software is about 1.5-2 years, the current search was limited to the last 5 years. The aim of this systematic review was not to provide a historical overview of AI technology in dentistry in general. Rather, the manuscript focused on prosthetic AI applications. Furthermore, the systematic search focused on clinical trials and case series with at least 5 patients to increase the scientific level (and to avoid including technical reports.”). In addition, we updated the reference list and included up-to-date publications in the Discussion.
Additional comments on the tables and figures
The fontsize of written statements in table 1 should be enlarged. The appendix of table can be shifted to the results section as another table.
The classification of different AI methods in Figure 1 is too generic, please consider supplementing more specific details; if better diagram other than Vien diagram can be used, please replace the current format. Meanwhile, the proposed workflow as charted in Figure 3 is also quite generic, which overlaps with some of your narrative statements in Section 3.1.
RESPONSE
The font size of Tables 1-3 has been modified and enlarged in order to guarantee a reader friendly version. You are absolutely right: the formerly Figure 1 presented a very superficial overview of AI; therefore, the text passage was extended and the old Figure 1 was deleted. The previous “Appendix Table” has been relocated to the section M&M because it described the methodological approach. Now Figure 2 (formerly Figure 3) followed the requirements of the guideline for Systematic Reviews (PRISMA). This format is routinely used in dental journals to summarize the search process. Therefore, we suggest keeping the mentioned workflow in Figure 1.
Round 2
Reviewer 1 Report
Thanks, I have no other questions.
Author Response
Thank you very much for your time and effort to review the manuscript.
Reviewer 2 Report
All the comments have been addressed correctly.
Author Response
Thank you very much for your time and effort to help improving the quality of the manuscript.
Reviewer 3 Report
Dear Authors,
Thanks so much for your updated manuscript on submission to Sensors Editorial Office. After my second round of review, I feel that this paper requires another round of revision, despite that some of the previous issues had been addressed. The current problematic issues are listed as follows:
a) This updated sysematic review paper had a focus on recent progress in AI applications to Prosthodontics. The research topic contains some interesting points and technical orientations are condensed. The listed tables and figures established related keynote work and the specific aspects in a small branch of artificial intelligence (AI) and its applications.
b) While some parts of this paper were edited and improved, the technical approach still looks generic, and lacks supportive quantitative results as well as strong evidence to support the systematic review. Previously, as I specified that "a few set of current evaluation parameters (study design, patients, training and test datasets) actually makes little sense to the topic itself. Hence, reviewers and potential readers expect to see your specific improvements on enabling the involved methodology (AI-based techniques, case studies, recent progress and applications) to at least relatively larger set of evaluation rubrics", please focus on this point and improve that. Thanks.
c) Another issues is the organization of this paper. The Section 1 missed a short summary on your actual contributions, and it is also absent from the last paragraph to specify the structure of rest sections (i.e., the remainder of this paper is structured as follows....) As I suggested before, a better version should include a section on the Related Work, while the authors skipped both of them in the first round of editing.
d) I assume that the Conclusion section has been significantly improved, which looks much better in contrast to prior version.
e) The tables and figures are limited, which propagates insufficient information to support the systematic review; meanwhile, I think the titles of each table, should be placed in front of the tabulated results (unlike the title of each figure should be preserved in the current version).
f) OK, I accept your updates in the enhanced list of References, while the abbreviation issues, along with the Capital/Uncapital first-letter issues of cited journal names (i.e., [4], [15], [20], [22]-[25], [27], [31]-[33]), should be calibrated in the next version.
g) Use of English was improved, while a few minor grammatical issues still need to be fixed.
h) Some other minor issues to be addressed: i) I don't think all the terms in the keywords should be bracketed with abbreviations; ii) Make the tabulated outcome in Table 3 more concise; iii) the overall length issues as a systematic review paper, I recommend the authors to reconsider that, which is at your side on increasing the probablity of acceptance.
Thank you and we look forward to seeing your improvements and greater success in the near future. Good luck!
Best regards,
Yours faithfully,
Author Response
Reviewer: 3
Comments to the Author
Thanks so much for your updated manuscript on submission to Sensors Editorial Office. After my second round of review, I feel that this paper requires another round of revision, despite that some of the previous issues had been addressed. The current problematic issues are listed as follows:
- a) This updated sysematic review paper had a focus on recent progress in AI applications to Prosthodontics. The research topic contains some interesting points and technical orientations are condensed. The listed tables and figures established related keynote work and the specific aspects in a small branch of artificial intelligence (AI) and its applications.
RESPONSE
Thank you very much for your time to review the manuscript and to help improve the quality.
- b) While some parts of this paper were edited and improved, the technical approach still looks generic, and lacks supportive quantitative results as well as strong evidence to support the systematic review. Previously, as I specified that "a few set of current evaluation parameters (study design, patients, training and test datasets) actually makes little sense to the topic itself. Hence, reviewers and potential readers expect to see your specific improvements on enabling the involved methodology (AI-based techniques, case studies, recent progress and applications) to at least relatively larger set of evaluation rubrics", please focus on this point and improve that.
RESPONSE
Thank you very much. Table 3 summarizes the extracted data of the identified studies in the field of AI applications in prosthodontics based on the systematic search. We thoroughly studied these publications in order to present to the reader a condensed extraction of these findings – following a descriptive approach due to the heterogeneity of the studies in terms of their specific aims and research designs. However, it is not possible to add more to the table if the data does not give us more at the moment. In our view, it is very important to report on the different topics in prosthodontics, which are in touch with AI technology today (although the number of studies identified is smaller than expected). In this context, it has to be seen as a motivation and a wake-up call for future research (and support from MedTech industry) in the field of AI in prosthodontics.
- c) Another issues is the organization of this paper. The Section 1 missed a short summary on your actual contributions, and it is also absent from the last paragraph to specify the structure of rest sections (i.e., the remainder of this paper is structured as follows....) As I suggested before, a better version should include a section on the Related Work, while the authors skipped both of them in the first round of editing.
RESPONSE
Thank you very much for this comment. We are happy to follow your suggestion and have added a section at the end of the Introduction to improve the readability.
- d) I assume that the Conclusion section has been significantly improved, which looks much better in contrast to prior version.
RESPONSE
Thank you very much. In the second revision, we worked on a few points again and made them clearer.
- e) The tables and figures are limited, which propagates insufficient information to support the systematic review; meanwhile, I think the titles of each table, should be placed in front of the tabulated results (unlike the title of each figure should be preserved in the current version).
RESPONSE
Thanks for your comment. We followed the given structure of the template of the journal SENSORS. If the editor agrees, we are happy to adjust the layout accordingly.
- f) OK, I accept your updates in the enhanced list of References, while the abbreviation issues, along with the Capital/Uncapital first-letter issues of cited journal names (i.e., [4], [15], [20], [22]-[25], [27], [31]-[33]), should be calibrated in the next version.
RESPONSE
Thanks for your help to improve the style of the manuscript. We checked the reference list and modified the Endnote-File in order to harmonize the capital letters of the first-letters of the journals.
- g) Use of English was improved, while a few minor grammatical issues still need to be fixed.
RESPONSE
Thanks for your comment. The entire manuscript was double-checked and proofread by a professional medical editor (UK-native).
- h) Some other minor issues to be addressed: i) I don't think all the terms in the keywords should be bracketed with abbreviations; ii) Make the tabulated outcome in Table 3 more concise; iii) the overall length issues as a systematic review paper, I recommend the authors to reconsider that, which is at your side on increasing the probablity of acceptance.
RESPONSE
Thank you very much for your valuable comments. We modified the Keywords and Table 3 accordingly.